# Effects of Heat Stress on the Laying Performance, Egg Quality, and Physiological Response of Laying Hens

**DOI:** 10.3390/ani14071076

**Published:** 2024-04-02

**Authors:** Hye-Ran Kim, Chaehwa Ryu, Sung-Dae Lee, Jin-Ho Cho, Hwanku Kang

**Affiliations:** 1Animal Nutrition and Physiology Division, National Institute of Animal Science, Rural Development Administration, Wanju 55365, Republic of Korea; ococ1004@korea.kr (H.-R.K.); chryu0629@korea.kr (C.R.); leesd@korea.kr (S.-D.L.); 2Department of Animal Science, Chungbuk National University, Cheongju 28644, Republic of Korea; jinhcho@chungbuk.ac.kr

**Keywords:** heat stress, poultry, temperature–humidity index, laying performance, physiological response

## Abstract

**Simple Summary:**

Climate change poses a significant challenge to the global livestock industry, intensifying weather conditions and exposing animals to extreme heat and humidity. Particularly in environments characterized by elevated temperatures and humidity, the thermoregulatory capacity of poultry is markedly diminished, thereby magnifying the deleterious effects of heat stress on their productive performance. Such climatic adversities not only impair the physiological homeostasis of chickens but also precipitate notable declines in production efficacy. Therefore, this study was conducted to investigate the impact of heat stress on laying hens utilizing the temperature–humidity index, with a focus on their laying performance, physiological responses, egg quality, and blood profiles under varying conditions of heat stress for 28 days. The research shows that heat stress significantly reduced feed intake and egg production and negatively affected egg quality, with changes observed in the hens’ blood profiles, such as altered levels of key electrolytes. These findings underscore the critical need for strategies to manage heat stress in poultry, highlighting the importance of understanding how environmental stressors affect poultry health and productivity. This study’s insights are valuable for improving livestock management practices, aiming to mitigate the adverse effects of climate change on the agriculture industry.

**Abstract:**

As high temperature and relative humidity (RH) are the main environmental factors causing heat stress, the temperature–humidity index (THI) serves as an indicator of heat stress in livestock animals. This study aimed to determine the effects of heat stress on the laying performance, physiological responses, egg quality, and blood profile of laying hens by subjecting them to environmental conditions with varying THI levels (68–85) for 28 days. The indicators of laying performance, such as feed intake (−30%) and egg production rate (−11%), significantly decreased in the hens exposed to severe heat stress (33 °C, 66% RH) compared to those exposed to thermoneutral conditions (21 °C, 68% RH). Moreover, severe heat stress reduced the egg yolk color, eggshell thickness and strength, and Haugh units of the eggs produced by the laying hens. Furthermore, a significant increase in serum K^+^ and a decrease in Na^+^ levels were observed in the hens subjected to severe heat stress compared with those under thermoneutral conditions. Our results indicate that heat stress alters the physiological responses and metabolism of laying hens, resulting in a lower egg quality and production rate.

## 1. Introduction

Climate change has become a significant challenge for the livestock industry worldwide by intensifying weather conditions that elevate ambient temperature and humidity [1]. Animals exposed to high ambient temperatures and humidity have a decreased ability to dissipate heat gained from the environment and generated from metabolic processes, making them susceptible to heat stress [2]. Several studies have reported that heat stress reduces the feed intake, body weight, egg production, fertility, and survivability of laying hens [3,4,5,6,7]. Mashaly et al. [3] reported that 31-week-old laying hens exposed to heat stress for 5 weeks had a significantly lower body weight, feed intake, egg production rate, and egg weight. Moreover, laying hens exposed to heat stress exhibit poorer egg quality [3,8,9].

Sweat glands aid in regulating the body temperature and in decreasing the heat load of most animals, as sweat evaporates from the skin. However, chickens lack sweat glands; instead, they are covered with feathers, which provide insulation and limit their ability to dissipate heat [10]. Consequently, chickens rely heavily on evaporative cooling through their mouth and respiratory tract as their primary means of thermoregulation; although effective, this adaptation renders them susceptible to high temperatures and humidity [1,11,12]. The temperature–humidity index (THI) is an environmental indicator commonly used to predict production losses incurred when livestock animals are exposed to hot and humid climatic conditions [13]. Generally, the thermoneutral zone of chickens ranges from 19 to 22 °C [12]; within this range, sensible heat loss is adequate to maintain the normal body temperature (40.6–42.4 °C) of chickens [14]. Beyond this range, chickens begin to exhibit the primary symptoms of heat stress: panting, prostrating, wing spreading, and consuming large amounts of water [15]. Panting, which refers to shallow and open-mouth breathing, increases the evaporation of water from the respiratory tract. When panting fails to maintain the body temperature of chickens, they become restless, lay comatose, and may even die [16].

Hyperventilation results in an excessive loss of CO_2_ from the lungs and blood of chickens [17]. Decreased CO_2_ levels induce elevated blood pH, causing the body to become alkaline—a condition known as respiratory alkalosis [16,18]. Changes in blood pH during respiratory alkalosis result in a reduced feed intake, lower overall performance, and decreased carbonic anhydrase activity, a critical enzyme for eggshell formation in laying hens [19]. Consequently, the shell gland secretes lower concentrations of calcium and carbonate, resulting in thin, weak eggshells [16]. Furthermore, during alkalosis, birds experience an acid–base disturbance characterized by lowered blood CO_2_ and H^+^ levels in response to heat stress [18,20]. Monovalent electrolytes (e.g., Na^+^, K^+^, and Cl^−^) are the key minerals involved in osmotic and acid–base balance, which is also considered as electrolyte balance. Recent studies have shown that broilers exposed to heat stress had decreased blood K^+^ and Na^+^ levels and increased Cl^−^ levels, which resulted in low electrolyte balance in the blood [15,21]. Such an imbalance disrupts the acid–base balance, thereby affecting several metabolic functions and the overall performance of chickens [22].

Biochemical changes in the blood reflect the physiological dynamics of organ systems and provide information on the health status and immune competence of animals [23]. Several studies have reported that heat stress affects hematological parameters, such as blood cell count [24,25,26]. Chronic heat stress causes a reduction in white blood cell (WBC) count and an increase in the heterophil/lymphocyte ratio [3,27]. Due to its high polyunsaturated acid content [23], the membrane of red blood cells (RBCs) is vulnerable to oxidative stress. Heat stress induces oxidative stress by causing the overproduction of reactive oxygen species (ROS), which exceeds the available antioxidant capacity of animal cells [28]; thus, the membrane of RBCs is easily damaged by heat stress [29]. Potentially toxic radicals and metabolic substances (e.g., ROS) are highly reactive and modify several biomolecules, such as proteins, lipids, and DNA [30], leading to several metabolic dysfunctions, including cell damage and death [28].

The aim of this study was to determine the interplay between physiological and biochemical changes in laying hens under environmental conditions with THI levels ranging from the alert to emergency zones [31] and to correlate these changes with the productivity and overall health of the hens.

## 2. Materials and Methods

### 2.1. Animals and Experimental Design

A total of 150 laying hens (26 weeks old, Lohmann Brown) were raised under thermoneutral conditions (21 °C and 60% relative humidity) for 14 days during the adaptation period, and the experimental treatments lasted for 28 days. The hens were divided randomly into five equal groups, each assigned based on the average laying rate (98.5%), that were maintained in a controlled chamber with THI values ranging from 68 to 85. Hens were housed in individual cages (300 × 370 × 570 mm; length × depth × height), equipped with nipples and a trough feeder, resulting in a total of thirty replicates across all treatment groups. All experiments were performed in a sustained manner. During the experimental period, the hens were provided with ad libitum access to water and food; they were fed with a commercial standard diet that met their nutritional requirements.

The experimental temperature and humidity were set according to the THI for laying hens [31]: comfort (THI < 70), alert (70 < THI < 75), danger (76 < THI < 81), and emergency (THI > 81). The THI was calculated based on the formula developed by the NRC (1971) using temperature and relative humidity (RH). Five treatment groups with corresponding temperature and RH were established: (1) THI 68 (21.0 ± 1.00 °C and 68.5 ± 6.15% RH), (2) THI 72 (24.6 ± 0.28 °C and 58.4 ± 2.96% RH), (3) THI 78 (28.5 ± 0.08 °C and 62.9 ± 3.60% RH), (4) THI 81 (28.9 ± 0.99 °C and 81.6 ± 6.32% RH), and (5) THI 85 (32.7 ± 0.26 °C and 66.1 ± 4.22% RH). Temperature and humidity loggers (174H; Testo^©^, Sparta, NJ, USA) were placed in each chamber to measure the temperature and RH every 30 min.

All experimental procedures performed in this study were reviewed and approved by the Institutional Animal Care and Use Committee of the National Institute of Animal Science (NIAS2020-0429).

### 2.2. Performance and Physiology Parameters

The body weight (BW) of each hen was recorded before (after the adaptation period) and after the 28-day experimental period. The number of eggs produced per cage was recorded daily. The weight of the eggs produced and amount of feed consumed by each hen were recorded daily and weekly. The feed conversion ratio (FCR) in each treatment group was calculated every week; the FCR was expressed as g of feed consumed per g of eggs produced. On Days 0, 7, 14, 21, and 28, 10 hens per treatment were randomly selected, and their daily water intake, rectal temperature, and respiration rate were measured. Rectal temperature was measured by inserting a rectal thermometer 3 cm deep into the rectum of each hen. Respiration rate per min was recorded by counting the breaths of the hens using a stopwatch.

### 2.3. Egg Quality

Thirty eggs per treatment were randomly collected on Days 7, 14, 21, and 28 and subjected to egg quality analysis. Each egg was weighed and then broken open for quality analysis. Eggshell strength (kgf/cm^2^) was measured using a rheometer (500DX; Sun Scientific, Tokyo, Japan), and eggshell thickness (mm) was measured using QCT (Miyutoyo, Kanagawa, Japan). The height of the thick albumen was measured using a QCM+ system (TSS, York, England), and yolk color was measured using COLOR METER (QCC) connected to the QCM+ system. The albumen height and egg weight measurements were used in Equation 1 to calculate the Haugh unit (HU) values:$$HU=100\log\left(H-1.7W^{0.37}+7.6\right)$$
where H is the albumen height (mm) and W is the egg weight (g).

### 2.4. Blood Sampling and Measurements

Ten birds per treatment were randomly selected for blood sampling at the end of the experiment. Blood was collected from the wing vein into EDTA and serum-separating tubes, with the procedure taking less than 5 min per bird and yielding a total volume of 3 mL. An automatic blood analyzer (BC-5300Vet; Mindray, Hamburg, Germany) was used to analyze the blood cell count consisting of WBC, RBC, neutrophil (NEU), lymphocyte (LYM), monocyte (MON), eosinophils (EOS), basophils (BAS), hemoglobin (HGB), hematocrit (HCT), and platelets (PLT); mean corpuscular volume (MCV); mean corpuscular hemoglobin (MCH); mean corpuscular hemoglobin concentration (MCHC); and red cell distribution width of the EDTA-anticoagulated blood samples. The serum electrolytes (Na^+^, K^+^, and Cl^−^) were measured using i-Smart 300^®^ (i-SENS, Seoul, Republic of Korea) based on the principle of ion-selective electrodes.

### 2.5. Feather Sampling and Corticosterone Measurement

Feather samples obtained from the wings of 10 laying hens per treatment were used to measure the corticosterone levels of the hens at the end of the experiment. The calamus of the plucked feathers was removed [32]; then, the feathers were pulverized using a bead beater at 50 Hz (taco^™^ Prep Bead Beater; GeneReach Biotechnology, Taichung, Taiwan). Subsequently, the samples were transferred into new polypropylene tubes containing 7 mL of methanol; then, the samples were placed in a shaking water bath at 50 °C for 15 h. After separating the methanol from the feather samples using a filtration funnel (Whatman^®^ filter paper), the feather samples were rinsed with an additional 2.5 mL of methanol. The combined extracts were dried at 50 °C and stored at −20 °C. For the corticosterone analysis, the dried extracts were thawed and mixed with the assay diluent (0.5 mL) from an enzyme immunoassay kit (Prod. No. K014-H; Arbor Assay, Ann Arbor, MI, USA), vortexed, and centrifuged at 1500× *g* for 15 min. To ensure the accuracy of the results, the samples were duplicated into two wells; as recommended by the kit, the optical density of the supernatant (50 µL) was measured using a microplate reader (SpectraMax^®^ Absorbance Readers, Molecular Devices, San Jose, CA, USA).

### 2.6. Statistical Analysis

The experiment was designed as a completely randomized design (CRD) with five treatment groups based on the THI levels, each containing 30 replicates per treatment, reflecting the individual hens housed in metabolic cages. Data are reported as mean ± standard error of the mean (SEM). All data were processed using the generalized linear matrix procedure in SAS version 9.4 (SAS 2009). Statistical differences among the treatment groups were determined by performing using Duncan’s multiple range test (SAS, 2009). A probability of *p* < 0.05 was considered significant, and 0.05 ≤ *p* ≤ 0.10 was considered a tendency.

## 3. Results and Discussion

### 3.1. Laying Performance

Varying THI levels significantly affected the laying performance of the hens (Table 1). The initial egg production of all treatment groups was similar. During the 4-week experimental period, except on Day 7, the egg production of the hens exposed to severe heat stress (THI = 85) was significantly lower (*p* < 0.01) than that of the other four groups. Compared with the THI 68 group, the feed intake (up to 15%; day) of the THI 81 group significantly decreased (*p* < 0.01) on Days 7, 14, and 21; meanwhile, the feed intake (up to 30%; d) of the THI 85 group significantly decreased (*p* < 0.01) throughout the experimental period. The FCR was significantly decreased (*p* < 0.01) with increasing THI levels on days 7 and 14. Heat stress also significantly affected (*p* < 0.05) the BW of the laying hens. The THI 68 and THI 72 groups slightly gained weight, whereas the hens in the THI 81 and THI 85 groups lost BW (*p* < 0.01) during the experimental period, except on Days 21 to 28 (*p* > 0.05).

### 3.2. Physiological Response

Table 2 shows the physiological responses of the laying hens to heat stress. The water consumption of the THI 85 group significantly increased (*p* < 0.05) compared with that of the THI 68 and THI 72 groups throughout the experimental period. Meanwhile, the rectal temperature of the THI 85 group was significantly higher (*p* < 0.05) than that of the THI 68 group on Days 7, 14, and 28. However, no significant difference in respiration rate was observed among the THI groups.

### 3.3. Egg Quality

All egg quality parameters were affected by heat stress (Table 3). However, no significant differences were observed between the THI 68 and THI 72 groups, except for the egg yolk color on Days 7 and 14. The HU was significantly lower (*p* < 0.01) in the THI 85 group compared to the other groups, except on Day 28. Compared with the THI 68 group, the egg yolk color significantly decreased (*p* < 0.01) in the THI 81 group on Days 7, 14, and 21 and in the THI 85 group at all time points. In the 4-week experiment period, the eggshell thickness and strength in the THI 85 group were significantly lower (*p* < 0.01) than those in the THI 68, 72, and 78 groups, except on Day 7 (*p* > 0.05), when only one trend was evident.

### 3.4. Blood Parameters

The effects of heat stress on the hematological parameters of the laying hens are shown in Figure 1. After 4 weeks of heat exposure, the RBC and HGB levels in the THI 85 group significantly decreased (*p* < 0.01) compared with those in the THI 68, 72, and 78 groups. The MCHC of the THI 81 and 85 groups also declined (*p* < 0.05) compared with that in the THI 68, 72, and 78 groups. However, no differences in WBC, NEU, LYM, MON, EOS, BAS, HCT, MCV, MCH, and PLT were observed among the treatment groups (*p* > 0.05). As shown in Table 4, serum K^+^ concentration significantly increased (*p* < 0.05) and Na^+^ concentration decreased in the THI 85 group compared with those in the THI 68 group (*p* < 0.05).

### 3.5. Corticosterone in Feathers

As shown in Table 5, the feather length and corticosterone concentration of laying hens did not differ significantly (*p* > 0.05) among the treatments.

## 4. Discussion

At high environmental temperatures, birds expend energy to regulate their body temperature and metabolic activity; hence, the energy intended for growth is diverted to the maintenance of homeostasis, resulting in performance loss. Several studies have reported that laying hens under hot environmental conditions exhibit decreased feed intake, egg production, feed efficiency, and BW [3,7,33]. RH is also a heat stress factor that influences the performance of chickens at high ambient temperatures [34]. Although Winn and Godfrey [35] defined RH greater than 80% when the temperature rises above 27 °C as ‘too wet’ to affect livestock performance, the contribution of RH to thermal stress in laying hens remains unknown and requires further research [7]. Hence, we assessed the effects of THI levels ranging from alert to emergency zones, as described by Zulovich and Deshazer [31], on the laying performance, egg quality, and physiological stress responses of laying hens. Our results revealed that heat stress severely affected the egg production of the hens throughout the 4-week experimental period. Egg production was significantly lower in the hens exposed to severe heat stress (THI = 85) than in those exposed to thermoneutral conditions (THI = 68). These findings are consistent with those of Mashaly et al. [3], who reported that the BW, feed consumption, and egg production and weight of 31-week-old laying hens were significantly reduced after exposure to heat stress for 5 weeks. A recent study found that the energy content of egg production was 61.8% of the energy consumed by control ducks and 63.4% in heat-stressed ducks, showing that reduced egg production did not fully compensate for the reduced energy intake, and even less so for the remaining energy [33]. In addition, we observed that the feed intake of the THI 81 (29 °C and 80% RH) and THI 85 (33 °C and 60% RH) groups significantly decreased compared with that of the THI 68 group for most of the experimental period; however, there were no significant differences in feed intake between the THI 79 (29 °C and 60% RH) and THI 68 groups. These results indicate that the difference between 60% and 80% RH at an environmental temperature of 29 °C affects the performance of laying hens. Similarly, Kim et al. [7] reported that the feed intake of laying hens in the THI 82 group (30 °C, 75% RH) significantly decreased compared with those in the THI 74 group (30 °C, 25% RH). Therefore, heat stress adversely affects egg production, and high RH and temperature affect the performance of laying hens. Under thermoneutral conditions, the core body temperature of mature birds is 41 °C; hence, birds utilize a wide temperature gradient for thermoregulation, facilitating heat loss [36]. Despite the well-documented impact of heat stress leading to an increased FCR [37], our study observed an anomaly, with a decrease in FCR among the groups subjected to heat stress. This unexpected result might be attributed to the extreme reduction in feed intake and efficient utilization of the ingested nutrients [38]. This observation suggests that employing pair-feeding methodologies in future research is essential for accurately measuring FCR under environmental stresses.

As the body temperature of birds rises accordingly with high ambient temperatures, their water intake and respiration rate significantly increase to lower their body temperature [15,39]. If the water deficit or the effect of evaporative cooling decreases due to high RH, heat stress may become more severe and cause death to birds [14]. Our results revealed that the water intake of the hens increased (*p* < 0.05) with increasing THI throughout 4 weeks. These results are consistent with those of Yahav et al. [40], who observed that the water consumption of young laying hens (8 to 10 months) increased with increasing ambient temperature. Meanwhile, the rectal temperature of the hens exposed to severe heat stress (THI = 85) was significantly higher than that of the hens under thermoneutral conditions (THI = 68), which is consistent with the findings of Chang et al. [41]. However, there was no significant difference in respiration rate, measured through the palpation method, among the THI groups. This result is contrary to the findings of Toyomizu et al. [20], where the respiratory rate of birds significantly increased under high ambient temperatures. The discrepancy between these results could be due to the different methods employed to measure the respiratory rate of birds: by palpation or visual inspection. The adverse effects of heat stress on the egg quality of laying hens have been well documented [3,7,8,9,33,39]. Regarding the decline in the reproductive performance of acutely heat-stressed hens, Mahmoud et al. [8] suggested that alterations in acid–base balance and Ca^2+^ levels and the diminished ability of duodenal cells to transport calcium could be critical factors influencing the detrimental effects of heat stress on the egg production, eggshell characteristics, and skeletal integrity of laying hens. In this study, the HU, egg yolk color, and eggshell thickness and strength of the hens exposed to severe heat stress (THI 85) significantly decreased, a trend consistently observed in all 4 weeks. These results are consistent with those of Allahverdi et al. [9], who found that the HU, an indicator of egg freshness, decreased in laying hens exposed to heat stress. Moreover, Mashaly et al. [3] and Lin et al. [39] reported that eggshell thickness and strength significantly decreased in heat-stressed laying hens. Similarly, Kim et al. [7] found that eggshell thickness and weight were lower at a high temperature and RH.

According to Rostagno [42], eggshell quality decreases due to lower feed intake, which reduces the amount of nutrients (e.g., Ca, Mg, and P) available for egg formation, and increased respiratory alkalosis, which reduces blood CO_2_ and HCO_3_ and elevates blood pH [5]; it reduces carbonic anhydrase levels in the shell gland and kidneys, and reduces calcium turnover [8]. These results indicate that the egg quality of laying hens decreased probably due to a significant reduction in feed intake, but no significant difference was observed in respiration rate. As the biochemical profile of blood is greatly affected by heat stress, it is often used as an indicator to assess the effect of heat stress on birds. Several studies have reported that heat stress reduces the RBC count and HGB levels in chickens [24,25,26]. Our results revealed that the RBC count, HGB levels, and MCHC of the hens exposed to heat stress (THI 85) significantly decreased after 4 weeks. According to Khan et al. [23], the basal metabolic rate of birds decreases under heat stress conditions; consequently, lower oxygen consumption leads to reduced hematopoietic activity and, thus, reduced RBC production. Increased water consumption and manure excretion due to heat stress result in increased electrolyte excretion; electrolytes play a major role in maintaining neuromuscular function, blood pH, and acid–base balance. In response to heat stress, hyperventilation in birds disrupts the acid–base balance, characterized by lowered blood CO_2_ and H^+^ concentrations [18,20,43]. These changes disrupt the mineral balance, leading to reduced production potential in heat-stressed chickens [44]; for instance, an increase in extracellular fluid pH results in a reduction in blood calcium levels, which, in turn, decreases calcium availability for eggshell production [44]. In this study, the serum Na^+^ levels significantly decreased and K^+^ levels significantly increased in the laying hens after 4 weeks of severe heat exposure (THI 85). The increase in plasma K^+^ levels and reduction in HCT values suggest cell damage with subsequent leakage of K^+^ out of the cells [20]. However, during alkalosis, extracellular K^+^ enters the cells and augments the secretion of K^+^ from the lumen of the renal tubule [22]; under these conditions, H^+^ is exchanged for K^+^ in the renal tubules. Increased K^+^ secretion results in reduced K^+^ concentrations in the blood, thereby causes circulation disorders in birds, leading to death [22]. Khone and Jones [45] demonstrated that heat stress increased the blood K^+^ levels in birds and suggested that it was related to the duration of exposure to heat stress.

Measuring corticosterone levels, which is a biological stress indicator, in samples such as blood, feces, and feathers is a common technique for monitoring the stress status of birds [46,47,48,49]. Blood sampling is the main method used to measure corticosterone levels as a stress biomarker in chickens. However, blood samples only provide a snapshot of corticosterone levels for 24 h prior to sampling, which represents acute stress and does not accurately represent chronic stress [46]. Several studies have reported the presence of corticosterone in chicken feathers as a putative biomarker of chronic stress [46,48,50]. According to Häffelin et al. [50], increased feather corticosterone concentrations (FCCs) can be measured after 1 week of treatment, considering the growth rate of feathers. Hence, FCC has been increasingly employed to measure chronic stress in chickens, which represents long-term stress over a period of weeks to months, and to provide an alternative matrix [32]. In this study, the use of pg/mm as a unit was following the recommendation of Bortolotti et al. [46] to investigate FCC within a single feather under consideration of its growth rate. The average FFC values of the five treatments ranged from 2.485 to 2.776 pg/mm, which is consistent with Bortolotti et al. [46], who reported that the total average value of corticosterone of every single sample, whether cut or plucked, was at 1.76 pg/mm for Anas and at 3.83 pg/mm for Answer. However, in this present study, the feather length (mm) and FCC (pg/mm) of the laying hens did not differ significantly among the THI treatments. The different results between this and other studies [32,46,50] that reported that chronic stress affects FCC in birds could be due to differences in the growth status of birds or the type of feather used. According to [50], various feather types grow over different periods at different rates. Moreover, knowledge on how age and sex can affect FCC remains inconsistent [51]. Thus, the results of this study may explain why corticosterone levels measured in the interscapular feathers of mature laying hens (26–30-week-old) are not affected by heat stress, and the application of FFC to laying hens requires consideration of feather growth and type.

## 5. Conclusions

The productivity indices of laying hens are significantly impacted by heat stress. Over a period of four weeks, hens exhibited reductions in egg production, feed intake, and egg quality due to chronic heat stress. Variations in egg quality, like eggshell thickness, could be attributed to reduced feed consumption. Furthermore, notable differences in blood parameters, such as red blood cells and electrolytes, are likely related to the increased water intake. Notably, our findings suggest that THI levels of 68 and 72 do not significantly impact laying hens, indicating a threshold for heat stress tolerance and recommending a re-evaluation of current heat stress mitigation strategies up to a THI of 72. Nevertheless, the extent of these impacts exhibits considerable variability, primarily attributed to the inherently complex and multifactorial nature of heat stress, coupled with the wide range of responses and adaptive mechanisms among birds within a cohort or population to such conditions. While specific indicators such as eggshell thickness pointed to the physiological adjustments to heat stress, they also revealed the necessity for standardized methodologies in assessing stress markers like respiratory rate and feather corticosterone concentration. Future studies should focus on developing a comprehensive framework for heat stress evaluation that incorporates a wider array of physiological, biochemical, and behavioral indicators. This approach will enable more nuanced understandings of heat stress responses and foster the development of targeted interventions to enhance poultry welfare and productivity in the face of climate change.

## Figures and Tables

**Figure 1 animals-14-01076-f001:**
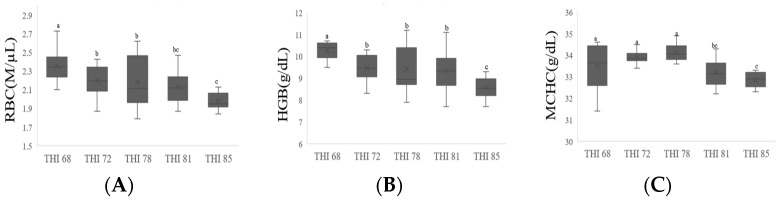
Effect of heat stress on red blood cells (**A**), hemoglobin (**B**), and mean corpuscular hemoglobin (**C**) of laying hens at 30 wks of age. ^a–c^ Different letters within figures differ significantly (*p* < 0.05. THI 68 (21 °C, 68% RH); THI 72 (25 °C, 58% RH); THI 78 (29 °C, 63% RH); THI 81 (29 °C, 82% RH); THI 85 (33 °C, 66% RH).

**Table 1 animals-14-01076-t001:** Effect of the temperature–humidity index (THI) on the performance of laying hens.

Item	Treatment ^1^					SEM ^2^	*p*-Value
	THI 68	THI 72	THI 78	THI 81	THI 85		
Egg production (%)							
Day 7	99.38	97.4	97.51	96.89	94.4	1.48	0.222
Day 14	97.51 ^a^	98.76 ^a^	98.81 ^a^	96.89 ^a^	92.85 ^b^	1.28	0.010
Day 21	96.27 ^a^	98.05 ^a^	98.81 ^a^	95.65 ^a^	85.70 ^b^	1.56	<0.001
Day 28	98.70 ^a^	96.89 ^a^	95.83 ^a^	95.03 ^a^	88.30 ^b^	1.79	0.002
Feed intake (g/day/bird)							
Day 7	129.5 ^a^	125.2 ^a^	120.3 ^ab^	110.6 ^b^	88.3 ^c^	4.26	<0.001
Day 14	127.5 ^a^	123.2 ^a^	119.0 ^a^	101.7 ^b^	91.2 ^c^	2.86	<0.001
Day 21	127.2 ^a^	118.2 ^ab^	115.4 ^b^	108.4 ^b^	81.4 ^c^	3.45	<0.001
Day 28	121.2 ^a^	122.0 ^a^	118.3 ^a^	114.2 ^a^	95.1 ^b^	3.12	<0.001
Egg weight (g)							
Day 7	58.0 ^a^	55.7 ^b^	55.5 ^b^	56.1 ^ab^	52.0 ^c^	0.68	<0.001
Day 14	57.8 ^a^	57.8 ^a^	57.2 ^a^	55.4 ^a^	48.9 ^b^	0.89	<0.001
Day 21	59.1 ^a^	56.7 ^b^	57.3 ^ab^	56.5 ^b^	50.1 ^c^	0.78	<0.001
Day 28	59.7 ^a^	57.3 ^ab^	55.8 ^bc^	53.6 ^c^	48.1 ^c^	0.87	<0.001
Feed conversion ratio (g/g)							
Day 7	2.40 ^a^	2.33 ^a^	2.08 ^b^	2.08 ^b^	1.78 ^c^	0.069	<0.001
Day 14	2.31 ^a^	2.23 ^ab^	2.06 ^bc^	1.82 ^d^	1.88 ^cd^	0.069	<0.001
Day 21	2.26	2.12	2.06	2.00	1.64	0.116	0.337
Day 28	2.19	2.23	2.18	2.07	2.12	0.061	0.353
Body weight gain (g/day/bird)							
Day 0 to 7	2.57 ^ab^	4.14 ^a^	2.43 ^ab^	−3.94 ^b^	−22.56 ^c^	2.48	<0.001
Day 7 to 14	7.07 ^a^	0.99 ^b^	0.27 ^b^	−0.53 ^a^	−8.68 ^c^	1.91	0.001
Day 14 to 21	4.32 ^a^	−1.95 ^ab^	−3.27 ^b^	−0.44 ^ab^	−12.57 ^c^	2.27	0.003
Day 21 to 28	2.05	4.39	−0.37	−0.76	−2.40	1.75	0.097
Day 0 to 28	15.62 ^a^	8.49 ^ab^	−0.94 ^bc^	−5.67 ^c^	−46.21 ^d^	4.05	<0.001

^1^ THI 68 (21 °C, 68% RH); THI 72 (25 °C, 58% RH); THI 78 (29 °C, 63% RH); THI 81 (29 °C, 82% RH); THI 85 (33 °C, 66% RH); ^2^ SEM, standard error of the means; ^a–d^ Different letters within rows differ significantly (*p* < 0.05).

**Table 2 animals-14-01076-t002:** Effect of heat stress on the water consumption, respiration rate, and rectal temperature of laying hens.

Items	Treatment ^1^					SEM ^2^	*p*-Value
	THI 68	THI 72	THI 78	THI 81	THI 85		
Water consumption (mL/day/bird)							
d7	202.1 ^b^	210.7 ^b^	308.8 ^a^	270.0 ^ab^	365.0 ^a^	30.84	0.005
d14	207.1 ^b^	164.3 ^b^	353.8 ^a^	270.6 ^ab^	325.4 ^a^	36.26	0.006
d21	202.1 ^c^	232.9 ^bc^	327.1 ^a^	293.8 ^ab^	283.1 a^b^	27.07	0.027
d28	192.9 ^b^	190.0 ^b^	263.8 ^ab^	205.0 ^ab^	285.0 ^a^	26.35	0.049
Respiration rate (breaths/min)							
d0	46.44	45.11	43.78	45.80	46.44	2.312	0.918
d7	48.44	46.40	48.40	47.40	48.29	3.060	0.988
d14	44.22	51.60	55.20	43.20	51.50	3.385	0.077
d21	50.89	49.40	64.80	53.80	58.44	4.977	0.205
d28	48.67	45.00	53.40	51.60	61.75	5.776	0.405
Rectal temperature (°C)							
d0	41.50	41.55	41.56	41.52	41.49	0.094	0.351
d7	41.61	41.54	41.85	41.79	41.80	0.083	0.050
d14	41.52 ^b^	41.61 ^b^	41.80 ^ab^	41.6 ^b^	41.99 ^a^	0.105	0.031
d21	41.56	41.59	41.75	41.72	41.68	0.076	0.340
d28	41.49 ^b^	41.60 ^b^	41.58 ^b^	41.68 ^b^	41.90 ^a^	0.065	0.002

^1^ THI 68 (21 °C, 68% RH); THI 72 (25 °C, 58% RH); THI 78 (29 °C, 63% RH); THI 81 (29 °C, 82% RH); THI 85 (33 °C, 66% RH); ^2^ SEM, standard error of the means; ^a–c^ Different letters within rows differ significantly (*p* < 0.05).

**Table 3 animals-14-01076-t003:** Effect of the temperature–humidity index on the egg quality of laying hens.

Item	Treatment ^1^					SEM ^2^	*p*-Value
	THI 68	THI 72	THI 78	THI 81	THI 85		
Haugh unit							
d7	99.29 ^a^	98.76 ^a^	97.13 ^a^	99.01 ^a^	92.09 ^b^	1.001	0.002
d14	100.2 ^a^	101.6 ^a^	100.14 ^a^	99.27 ^a^	93.59 ^b^	1.345	0.003
d21	99.69 ^a^	100.5 ^a^	97.92 ^ab^	101.1 ^a^	95.78 ^b^	1.109	0.015
d28	94.09	92.16	92.99	94.09	93.11	1.518	0.889
Egg yolk color							
d7	2.693 ^ab^	2.725 ^a^	2.817 ^a^	2.503 ^bc^	2.478 ^c^	0.068	0.006
d14	3.503 ^a^	3.197 ^b^	3.160 ^b^	3.047 ^b^	2.593 ^c^	0.104	<0.001
d21	3.868 ^a^	3.495 ^ab^	3.477 ^ab^	3.222 ^b^	2.660 ^c^	0.176	0.001
d28	3.460 ^a^	3.003 ^b^	3.208 ^ab^	2.918 ^b^	2.587 ^c^	0.110	<0.001
Eggshell thickness (mm)							
d7	0.388 ^a^	0.388 ^a^	0.370 ^a^	0.347 ^b^	0.327 ^c^	0.007	<0.001
d14	0.386 ^a^	0.378 ^a^	0.375 ^a^	0.352 ^b^	0.317 ^c^	0.008	<0.001
d21	0.402 ^a^	0.393 ^a^	0.368 ^a^	0.355 ^bc^	0.333 ^c^	0.008	<0.001
d28	0.390 ^a^	0.388 ^a^	0.365 ^b^	0.343 ^c^	0.322 ^d^	0.007	<0.001
Eggshell strength (kgf/cm^2^)							
d7	39.18 ^a^	40.49 ^a^	38.18 ^a^	32.80 ^b^	32.23 ^b^	1.130	<0.001
d14	41.53 ^a^	41.13 ^a^	39.22 ^a^	37.51 ^a^	30.41 ^b^	1.623	0.001
d21	43.29 ^ab^	44.01 ^a^	41.26 ^ab^	38.64 ^b^	35.33 ^c^	1.522	0.003
d28	41.98 ^a^	42.21 ^a^	41.15 ^a^	38.19 ^ab^	34.18 ^b^	1.435	0.003

^1^ THI 68 (21 °C, 68% RH); THI 72 (25 °C, 58% RH); THI 78 (29 °C, 63% RH); THI 81 (29 °C, 82% RH); THI 85 (33 °C, 66% RH); ^2^ SEM, standard error of the means; ^a–d^ Different letters within rows differ significantly (*p* < 0.05).

**Table 4 animals-14-01076-t004:** Effect of heat stress on concentration of serum electrolytes (Na^+^, K^+^, and Cl^−^) of laying hens.

Items	Treatment ^1^					SEM ^2^	*p*-Value
	THI 68	THI 72	THI 78	THI 81	THI 85		
Na (mmol/L)							
d14	159.5	166.5	162.5	171.0	163.3	4.166	0.426
d28	192.9 ^b^	190.0 ^b^	263.8 ^ab^	205.0 ^ab^	285.0 ^a^	26.35	0.049
K (mmol/L)							
d14	4.500 ^b^	4.733 ^b^	4.925 ^ab^	5.438 ^a^	4.943 ^ab^	0.177	0.021
d28	4.163 ^c^	4.400 ^bc^	4.600 ^b^	4.638 ^b^	5.038 ^a^	0.130	<0.001
Cl (mmol/L)							
d14	123.8	126.3	124.8	130.5	126.1	3.099	0.639
d28	115.1	115.6	114.4	115.5	114.0	0.829	0.575

^1^ THI 68 (21 °C, 68% RH); THI 72 (25 °C, 58% RH); THI 78 (29 °C, 63% RH); THI 81 (29 °C, 82% RH); THI 85 (33 °C, 66% RH); ^2^ SEM, standard error of the means; ^a–c^ Different letters within rows differ significantly (*p* < 0.05).

**Table 5 animals-14-01076-t005:** Effect of heat stress on corticosterone concentration in feathers of laying hens.

Items	Treatment ^1^					SEM ^3^	*p*-Value
	THI 68	THI 72	THI 78	THI 81	THI 85		
Feather length(mm)	126.5	139.4	133.6	137.4	139.5	3.132	0.2109
FCC ^2^ (pg/mm)	2.626	2.776	2.502	2.576	2.485	0.336	0.9187

^1^ THI 68 (21 °C, 68% RH); THI 72 (25 °C, 58% RH); THI 78 (29 °C, 63% RH); THI 81 (29 °C, 82% RH); THI 85 (33 °C, 66% RH). ^2^ FCC, feather corticosterone concentration; ^3^ SEM, standard error of the means.

## Data Availability

The data presented in this study are available upon request from the first author. The data are not publicly available because of restrictions imposed by the research group.

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
