# Peer review of "Effects of Heat Stress on the Laying Performance, Egg Quality, and Physiological Response of Laying Hens"

_animals, 2024, doi:10.3390/ani14071076_

Round 1
Reviewer 1 Report
Comments and Suggestions for Authors
Overall, the paper is an interesting read. The introduction sets up the study nicely and describes parameters that will be tested. However, the materials and methods is lacking much information on the basic animal housing and husbandry set up. It also does not confirm whether the assigned temperature and relative humidity were achieved with high precision. One would read this paper but would not be able to replicate the study, which is a serious flaw. Adding information would help.
Furthermore, the data collection section indicated many parameters measured, however several were not found in the results section, but were a discussion point in the discussion section. Discussion was insightful but should be split into several paragrahs.
Lastly, there is so much potential and conclusion to be drawn from the study, but the conclusion is short and diminishes these potentials. More details can be added to this paragraph.
Additional comments in the attached document.

Comments on the Quality of English Language
Be sure to write consistently in past tense.
Author Response
Thank you very much for your thorough review and valuable comments on our manuscript. In response, we have carefully considered each of your suggestions and made corresponding revisions to the document. These changes include clarifying the methodology, providing additional details on the study design, improving the presentation of our results, and reinforcing the support for our conclusions. The revised sections have been highlighted and tracked in the resubmitted files for easy identification. We believe these amendments significantly enhance the clarity, coherence, and overall quality of our manuscript, better aligning it with the journal's standards and expectations. We appreciate the opportunity to improve our work based on your insightful feedback and hope that the revisions adequately address your concerns.

Reviewer 2 Report
Comments and Suggestions for Authors
Dear Authors,
The introduction is too long, it could be a little shorter.
The characteristics given in the method section are not given in the results section.
An increase in respiratory rate is expected due to an increase in temperature. However, in your study, it appears that there is no change in respiratory rate. How do you explain this?
Materials and Methods
Have you obtained any data on viability?
Line 115: Please specify the hen genotype.
Lines 115-121: Please specify the stocking density and feed composition.
It should be stated how many replications the study consists of and how many animals there are in each replication.
Information should be given about whether the animals are kept in cages and about feeders and drinkers.
Lines 138-140: Please check this sentence “The feed conversion ratio (FCR) in each treatment group was calculated every 2 weeks; the FCR was expressed as kg of feed consumed per kg of eggs produced.”. Feed efficiency is stated weekly in the Table 1. Additionally, the unit is expressed as g/g.
Line 147: Please check “Each egg was weighed and cracked”, cracked?
Results
Lines 203-204: Please check this sentence “However, there was no significant difference (P > 0.05) in FCR among the THI groups.”
Table 1: Although the P value was significant in the feed evaluation on the seventh and 14th days, it was not specified with superscripts.
Table 1: It would be more appropriate to indicate body weight losses as a percentage.
Table 1: Check the feed evaluation calculations again. For example, you have stated the feed evaluation on day 7 as 2.40, but in the calculation you specified in the method (129.5/58=2.23), the FCR is calculated as 2.23.
Table 3: Although the measurement of egg yolk index is not specified in the method section, the Table is given.
Table 3: Please check the eggshell strength unit. It conflicts with the unit specified in the method.
Table 3: Although egg yolk color is given in the method section, it is not given in the Tables.
Lines 225-226: Please check this sentence “However, no significant differences were observed between the THI 68 and 72 groups.”
Lines226-228: Please check this sentence “The HU of the THI 85 group was significantly lower (P < 0.05) than that of the other four groups throughout the experimental period.”. In the last period, differences between groups in terms of Haugh unit appear to be insignificant.
Table 4: “As shown in Table 4,”. There is no table 4.
Blood cell count and corticosterone results are not given in the tables.
Discussion
Line 320: Please check this sentence “an increase in respiration rate;”.
Lines 337-338: These results are not included in Tables.
Conclusions
Which e temperature–humidity index group would you recommend based on the results of the study?
Line 381: Please check “an increased respiratory rate”

Author Response
Thank you very much for your thorough review and valuable comments on our manuscript. In response, we have carefully considered each of your suggestions and made corresponding revisions to the document. These changes include clarifying the methodology, providing additional details on the study design, improving the presentation of our results, and reinforcing the support for our conclusions. The revised sections have been highlighted and tracked in the resubmitted files for easy identification. We believe these amendments significantly enhance the clarity, coherence, and overall quality of our manuscript, better aligning it with the journal's standards and expectations. We appreciate the opportunity to improve our work based on your insightful feedback and hope that the revisions adequately address your concerns

Round 2
Reviewer 1 Report
Comments and Suggestions for Authors
This is much better than the first draft. I still have a few additional comments (see attached), but great job and great work in making the edits.

Comments on the Quality of English Language
Author Response
Thank you once again for your invaluable feedback on our manuscript. Following your latest comments, we have undertaken a second round of revisions to further refine and enhance our study. Please see the attachment. In our point-by-point response, we explained our methodology for measuring daily water intake, standardized statistical notation across tables and figures, unified unit presentation, detailed our approach to quantifying egg yolk color changes under heat stress, and corrected an editing error in Table. We trust that these adjustments further improve the manuscript, bringing it in closer alignment with the journal's standards. We are grateful for the opportunity to refine our work through your constructive critiques and are hopeful that our manuscript now meets your expectations for publication.

Reviewer 2 Report
Comments and Suggestions for Authors
Dear Authors,
Thank you for following the recommendations mentioned except for the FCR. Feed efficiency is an important criterion. I would like to state that I do not agree with your statements regarding the FCR. It may be more appropriate to give the feed efficiency as a decimal number.
Best regards,
Author Response
Thank you for your critical insight regarding our presentation and interpretation of the Feed Conversion Ratio (FCR) in our study. We wholeheartedly agree with your perspective on the significance of presenting feed efficiency accurately, especially given its importance as a criterion in poultry science research. Your comment has prompted us to reflect deeply on our findings and the manner in which we have conveyed them.
It is well-documented in the literature, as you pointed out with references to Sahin et al. (2002), Mashaly et al. (2004), and He et al. (2019), that heat stress typically leads to an increase in FCR due to reduced feed intake and potentially compromised egg production efficiency. Contrary to these established findings, our study observed a decrease in FCR under heat stress conditions, presenting an unusual scenario that diverges from conventional expectations.
We further analyzed the FCR considering the laying rate, presenting the results in grams per dozen eggs produced (table below). This detailed analysis demonstrates that, even when considering the laying rate and calculating FCR as grams per dozen eggs produced, the observed trend of decreased FCR under heat stress conditions remains consistent.
|
Item |
Treatment1 |
SEM2 |
P-value |
||||
|
THI 68 |
THI 72 |
THI 78 |
THI 81 |
THI 85 |
|||
|
Egg production (%) |
|||||||
|
Day 7 |
99.38 |
97.4 |
97.51 |
96.89 |
94.4 |
1.48 |
0.2219 |
|
Day 14 |
97.51a |
98.76a |
98.81a |
96.89a |
92.85b |
1.28 |
0.0095 |
|
Day 21 |
96.27a |
98.05a |
98.81a |
95.65a |
85.70b |
1.56 |
<.0001 |
|
Day 28 |
98.70a |
96.89a |
95.83a |
95.03a |
88.30b |
1.79 |
0.0021 |
|
Feed intake (g/day/bird) |
|||||||
|
Day 7 |
129.5a |
125.2a |
120.3ab |
110.6b |
88.3c |
4.26 |
<.0001 |
|
Day 14 |
127.5a |
123.2a |
119.0a |
101.7b |
91.2c |
2.86 |
<.0001 |
|
Day 21 |
127.2a |
118.2ab |
115.4b |
108.4b |
81.4c |
3.45 |
<.0001 |
|
Day 28 |
121.2a |
122.0a |
118.3a |
114.2a |
95.1b |
3.12 |
<.0001 |
|
Feed Conversion Ratio (g/dozen) |
|||||||
|
Day 7 |
1.55 a |
1.49 ab |
1.48 ab |
1.41b |
1.22 c |
0.039 |
<.0001 |
|
Day 14 |
1.54 a |
1.49 a |
1.44 a |
1.25 b |
1.22 b |
0.042 |
<.0001 |
|
Day 21 |
1.57 a |
1.45 ab |
1.42 b |
1.38 b |
1.13 c |
0.043 |
<.0001 |
|
Day 28 |
1.48 ab |
1.53 ab |
1.63 ab |
1.45 ab |
1.34 b |
0.07 |
<.0001 |
Upon further consideration, we speculate that this unexpected result could be attributed to a significant reduction in feed intake among the heat-stressed laying hens, which might have led to a highly efficient utilization of the consumed nutrients and an increased nutrient absorption rate. This speculation, while cautious, attempts to rationalize the observed decrease in FCR despite the adverse conditions.
Recognizing that our findings deviate from the norm and do not align with established theories on FCR under heat stress, we acknowledge the potential for our results to contribute to ongoing discussions in the field rather than definitively contradicting current understandings. To address this, we propose in our discussion section the potential necessity of employing a pair-feeding approach in future heat stress experiments (Lines 275-280).
We are committed to maintaining the integrity and reliability of our research. If deemed necessary to avoid confusion among readers, we are prepared to omit the FCR section from our manuscript. We hope that our cautious interpretation and proposed future research directions demonstrate our acknowledgment of the anomaly and our dedication to contributing valuable insights to the field.
Your feedback has been invaluable in helping us address this complex issue, and we are grateful for the opportunity to refine our work further.